# Genome-Wide Identification of the Litchi BBX Gene Family and Analysis of Its Potential Role in Pericarp Coloring

**DOI:** 10.3390/ijms262210834

**Published:** 2025-11-07

**Authors:** Tao Liu, Yanzhao Chen, Weinan Song, Hongna Zhang, Yongzan Wei

**Affiliations:** 1School of Life Sciences and Medicine, Shandong University of Technology, Zibo 255049, China; liutlsy@126.com (T.L.); chenyz335@163.com (Y.C.); 2Hainan Provincial Key Laboratory of Quality Control of Tropical Horticultural Crops, School of Tropical Agriculture and Forestry, Sanya Institute of Breeding and Multiplication, Hainan University, Haikou 570228, China; 18503483257@163.com (W.S.); 13692476979@139.com (H.Z.); 3National Key Laboratory for Tropical Crop Breeding, Institute of Tropical Bioscience and Biotechnology, Chinese Academy of Tropical Agricultural Sciences, Haikou 571101, China; 4Sanya Research Academy, Chinese Academy of Tropical Agriculture Science, Sanya 572019, China

**Keywords:** *Litchi chinensis* Sonn., B-Box protein, anthocyanins, gene expression, light responsive

## Abstract

Litchi is an important subtropical fruit, highly valued by consumers for its vibrant color and distinctive flavor. B-box (BBX) proteins, which are zinc finger transcription factors, play a crucial role in regulating plant growth, development, and stress responses. Nevertheless, the specific function of *BBX* genes in the development and coloration of litchi fruit remains inadequately understood. In this study, 21 *LcBBX* genes (designated as *LcBBX1*-*LcBBX21*) were identified within the litchi genome. These genes were categorized into five sub-families based on phylogenetic analysis and were found to be unevenly distributed across 12 chromosomes. Promoter analysis revealed a rich presence of light-responsive elements, such as the G-box, and abscisic acid (ABA) responsive elements, including ABRE, within the promoter regions of *LcBBX* genes. Protein–protein interaction predictions indicated that the majority of *LcBBX* genes have the potential to interact with the light-responsive factor HY5. Transcriptome analysis and qRT-PCR results demonstrated that *LcBBX* genes exhibit tissue-specific expression patterns. Notably, most *LcBBX* genes were highly expressed prior to fruit coloration, whereas *LcBBX4* and *LcBBX10* were upregulated during the fruit coloration phase. Furthermore, *LcBBX1*/*4*/*6*/*7*/*15*/*19* were upregulated in response to light following the removal of shading. The findings suggest that *LcBBX4* may directly regulate anthocyanin biosynthesis in litchi pericarp. This study provides critical insights into the molecular mechanisms underlying litchi fruit development and coloration.

## 1. Introduction

Transcription factors (TFs) can regulate various processes in plant growth and development, as well as responses to biotic or abiotic stresses, by activating or inhibiting the expression of their target genes [1]. Zinc finger transcription factors are among the largest TF families in plants and play a crucial role in regulating diverse biological functions [2]. BBX proteins are a class of zinc finger transcription factors containing conserved B-box domains, which can interact with DNA, RNA, or proteins to precisely regulate plant growth and development [3]. BBX proteins exhibit typical structural features: they usually contain one or two B-box motifs at the N-terminus (each approximately 40 amino acid residues in length), and some members also have a CCT motif in the C-terminal region [4]. Based on the consensus sequence and spacing characteristics of zinc-binding residues, B-box domains can be divided into two types, namely B-box1 (C-X2-C-X7-8-C-X2-D-X-A-X-L-C-X2-C-D-X3-H) and B-box2 (C-X2-C-X3- P-X4-C-X2-D-X3-L-C-X2-C-D-X3-H) [5]. The CCT domain contains 42–43 amino acid residues, and previous studies have confirmed that this domain plays a key role in the DNA-binding ability, transcriptional activity, and nuclear localization function of BBX proteins [6,7]. According to the differences in the composition of B-box and CCT domains, the BBX protein family is usually divided into 5 subfamilies [8,9].

As an important class of regulatory factors in plants, BBX proteins play critical roles in multiple processes of plant growth and development, including the regulation of plant flowering, seedling photomorphogenesis, pigment accumulation, and abiotic stress response. *CONSTANS* (*AtBBX1*) is the first identified BBX protein in *Arabidopsis thaliana*; under long-day conditions, it can activate the expression of the *FLOWERING LOCUS* (*FT)* gene to regulate the photoperiodic flowering process [10]. In the flowering regulatory pathway, different BBX proteins exhibit distinct functions: *AtBBX6* and *AtBBX24* can promote plant flowering [11,12], while *AtBBX4*, *AtBBX5*, and *AtBBX7* exert flowering-inhibiting effects [13,14,15]. In addition, under red light conditions, AtBBX11 interacts with *PHYTOCHROME-INTERACTING FACTOR 3* (*PIF3*), thereby promoting photomorphogenesis [16]. In contrast, *AtBBX30* and *AtBBX32* are negative regulators of photomorphogenesis, and their expression at the transcriptional level is negatively regulated by *ELONGATED HYPOCOTYL 5* (*HY5*) [17].

*BBX* genes have been confirmed to be involved in regulating anthocyanin biosynthesis in multiple species, including pear [18], apple [19], cherry [20], strawberry [21], and grape [22]. In *Arabidopsis thaliana*, *AtBBX21*/*22*/*23* positively regulate anthocyanin accumulation [23,24,25], while *AtBBX24/25* exert negative regulation [26,27]; under UV-B irradiation, *AtBBX31* interacts with *HY5* to significantly promote anthocyanin accumulation [28]. In contrast, *AtBBX32* can negatively regulate anthocyanin biosynthesis by inhibiting the activity of the BBX21-HY5 complex [29]. In addition, BBX proteins also play important regulatory roles in abiotic stress responses. For example, heterologous expression of *AtBBX29* in sugarcane can significantly increase proline accumulation and antioxidant enzyme activity in sugarcane, thereby enhancing sugarcane’s drought resistance and anti-aging ability [30]; overexpression of *CmBBX24* in chrysanthemum markedly improves tolerance to freezing and drought stresses [31]; heterologous expression of apple *MdBBX10* in *Arabidopsis* significantly increases seed germination rate and seedling root length, while enhancing tolerance to drought and salt stresses [32].

Litchi (*Litchi chinensis* Sonn.), an evergreen tree of the genus Litchi (*Sapindaceae*), is an important subtropical fruit. The reddening of litchi pericarp is mainly driven by anthocyanin accumulation [33]. R2R3-MYB, bHLH, and WD40 are core transcription factor families that regulate anthocyanin synthesis [34]. Among them, the *LcMYB1* transcription factor in litchi has been confirmed to positively regulate the biosynthesis of pericarp anthocyanins [35,36,37,38]. Transcription factors such as *NAC* [39], *bZIP* [40], and *ERF* [41] have also been successively identified and proven to be involved in anthocyanin accumulation in litchi fruit. However, whether *BBX* transcription factors regulate anthocyanin accumulation in litchi pericarp remains unclear. To clarify the regulatory mechanism of *BBX* genes on anthocyanin biosynthesis in litchi pericarp, this study conducted a systematic genome-wide identification and analysis of the litchi *BBX* gene family, including analyses of physicochemical properties, phylogenetic and evolutionary relationships, chromosomal localization, collinearity, conserved domain characteristics, gene structure, promoter *cis*-acting element prediction, and protein–protein interaction (PPI) network prediction of *LcBBX* genes. Meanwhile, we comparatively analyzed the expression patterns of litchi *BBX* family members across different tissues, pericarp developmental stages (pre-coloration, coloration, ripening), and corresponding light conditions, screened candidate *LcBBX* genes potentially regulating litchi pericarp coloration, and provided a theoretical basis for further exploring the function of *LcBBX* genes in anthocyanin biosynthesis and pericarp coloration.

## 2. Results

### 2.1. Identification and Physicochemical Properties of LcBBX Genes

In this study, we identified 21 *BBX* family members from the litchi genome and designated them *LcBBX1* to *LcBBX21* based on their chromosomal order. Physicochemical analysis revealed that the amino acid lengths of *LcBBX* proteins range from 132 aa (*LcBBX16*) to 499 aa (*LcBBX19*), with molecular weights varying between 15.21 and 56.19 kDa. The predicted isoelectric points ranged from 4.44 (*LcBBX5*) to 9.67 (*LcBBX7*). Based on instability index predictions, only *LcBBX4*, *LcBBX11*, and *LcBBX21* were classified as stable proteins (instability index < 40), whereas the remaining members were unstable. The results of subcellular localization prediction showed that all *LcBBX* proteins are localized in the nucleus, except that *LcBBX2* is localized in chloroplasts and *LcBBX14* is localized in the cytoplasm (Appendix A).

### 2.2. Phylogenetic Analysis of LcBBX Genes

To elucidate the evolutionary relationships among the BBX genes of litchi, *Arabidopsis*, rice, pear, and pineapple, we constructed a phylogenetic tree using the BBX protein sequences from *Arabidopsis* (32), rice (30), pear (25), pineapple (19), and litchi (21). The cluster analysis revealed that the *LcBBX* genes could be classified into five distinct subfamilies (Group I–V), containing 4, 3, 2, 7, and 5 members, respectively. Previous studies have demonstrated that both *AtBBX21* and *AtBBX22* act as positive regulators of anthocyanin biosynthesis in *Arabidopsis* [23,24]. Among them, *LcBBX8* clustered closely with *AtBBX21*, and *LcBBX4* grouped with *AtBBX22*, suggesting that these BBX proteins may share a common evolutionary origin and could perform similar biological roles in litchi (Figure 1).

### 2.3. Conserved Domains, Gene Structure, and Conserved Motifs of the LcBBX Genes

Conserved domain analysis showed that all 21 *LcBBXs* in litchi contain the B-box domain, among which members of subfamilies Group I/II/III also contain the CCT domain (Figure 2B). Gene structure analysis revealed that the number of introns in *LcBBX* genes shows subfamily-specific characteristics. In subfamilies Group Ⅰ/Ⅲ/Ⅳ, all members contain only 1 intron except for *LcBBX7/16*, which have no introns, while *LcBBX* members in subfamily Group Ⅱ contain multiple introns (4–5 introns) (Figure 2C).

In the analysis of conserved motifs in litchi BBX protein sequences, 10 distinct conserved motifs were identified (Appendix A). Combined analysis of the conserved motif results (Appendix A) and the conserved domain results (Figure 2B) revealed that Motif1 corresponds to the Bbox1 domain, Motif3 corresponds to the Bbox2 domain, and Motif2 corresponds to the CCT domain. Additionally, Motif5/6/7/8 are exclusively present in *LcBBX20/21*, Motif9 is only found in *LcBBX5/12/15*, and Motif10 exists solely in *LcBBX13/18*.

### 2.4. Chromosomal Localization and Collinearity Analysis of LcBBX Genes

The 21 *LcBBX* genes were unevenly distributed across 12 litchi chromosomes. Among them, chromosomes 1 and 12 each contained 3 *LcBBX* genes; chromosomes 2, 5, 8, 14, and 15 each had 2 *LcBBX* members; and chromosomes 3, 4, 9, 10, and 11 each harbored 1 *LcBBX* gene. Further observation revealed that most *LcBBX* genes were located in the gene-dense regions of the chromosomes (Figure 3).

To explore the expansion mechanism of the *LcBBX* gene family in litchi, we performed an intraspecific collinearity analysis (Figure 3). The results showed that there were 4 segmental duplication events in the litchi *LcBBX* family, involving *LcBBX8-LcBBX10*, *LcBBX9-LcBBX14*, *LcBBX12-LcBBX15*, and *LcBBX13-LcBBX18*. In addition, *LcBBX20* and *LcBBX21* are a pair of tandem duplicated genes, with a sequence similarity of approximately 99.16%.

In addition, to further explore the evolutionary characteristics of the litchi *BBX* gene family, this study analyzed the collinearity relationships between litchi and related species (longan, rambutan), important fruit trees (apple, pineapple), and model plants (*Arabidopsis*, rice) (Figure 4). Among them, longan and rambutan, which belong to the same *Sapindaceae* family as litchi, share 23 and 26 homologous *BBX* genes with litchi, respectively (Figure 4A); apple and *Arabidopsis*, as dicotyledonous plants like litchi, share 41 and 25 homologous *BBX* genes with litchi, respectively (Figure 4B); pineapple and rice are monocotyledonous plants, and they have relatively fewer homologous *BBX* genes with litchi, with only 10 and 4, respectively (Figure 4C).

### 2.5. Analysis of cis-Acting Elements in the Promoters of LcBBX Genes

Analyzing and predicting the type, number, and location of *cis*-acting elements in the promoter of a target gene is of great significance for revealing the molecular basis of how genes are precisely regulated. The *cis*-acting elements in the promoter region of litchi *LcBBX* genes were predicted and analyzed using the PlantCARE database. The results showed that the promoters of all members of the *LcBBX* family contained a variety of *cis*-acting elements, mainly involving elements related to light response, hormone response, stress response, and growth and development regulation (Figure 5). Among these, the light response elements had the largest number of types and quantities, with the G-Box element accounting for the highest proportion (84 in total); these light response elements were unevenly distributed in the *LcBBX* promoter sequences (Appendix A), indicating that the expression of litchi *BBX* family genes may be regulated by light signals.

In the promoter sequences of *LcBBX* genes, a variety of *cis*-acting elements related to endogenous hormone responses were also predicted, including auxin-responsive *cis*-acting elements (TGA-element and AuxRR-core), ABA-responsive *cis*-acting element (ABRE), gibberellin-responsive *cis*-acting elements (TATC-box, GARE-motif, and P-box), methyl jasmonate-responsive *cis*-acting element (CGTCA-motif), and salicylic acid-responsive *cis*-acting element (TCA-element) (Figure 5). Among them, except for *LcBBX13*, the promoter regions of all other members contain at least one ABRE element, indicating that ABA plays an important role in the transcriptional regulation of *LcBBX*.

A large number of stress-responsive *cis*-acting elements were also identified in the promoter sequences of *LcBBX* genes. Among them, the anaerobic induction element (ARE) was present in the promoter sequences of 19 *LcBBX* genes; the drought-induced response element (MBS) was found in 13 *LcBBX* genes; the low-temperature-induced response element (LTR) existed in 4 *LcBBX* genes; and the defense and stress response element (TC-rich repeats) was detected in 6 *LcBBX* genes. In addition, elements such as the zein metabolism regulatory element (O2-site), meristem expression regulatory element (CAT-box), and circadian rhythm regulatory element (circadian) were also present in the promoter sequences of multiple *LcBBX* genes. These results indicate that *LcBBXs* play important roles in plant growth and development as well as in responses to external stresses.

### 2.6. GO Enrichment Annotation and Protein–Protein Interaction Network Prediction of LcBBX Genes

GO functional enrichment analysis was performed on 21 *LcBBX* genes (Appendix A), which could be classified into three categories: 16 biological processes, 4 molecular functions, and 1 cellular component (Appendix A). In the biological process category, *LcBBX* genes were significantly enriched in terms related to "response to light" and "post-embryonic development", indicating that *LcBBX* genes may be involved in plant light signal response and growth and development regulation. In the molecular function category, the functions of *LcBBX* genes were mainly associated with "DNA-binding transcription factor activity", which can bind to the transcriptional *cis*-regulatory regions of DNA molecules, thereby activating or inhibiting the expression of downstream structural genes. In the cellular component category, *LcBBX* genes were enriched in the nucleus, suggesting that they may be involved in transcriptional regulation processes within the nucleus.

The results of protein interaction prediction showed that there are numerous interaction relationships among members of the litchi *BBX* genes, with the number of interactions ranging from 1 to 10. Among them, LcBBX19 may interact with 10 other LcBBX proteins. The light response factor HY5 and its homologous protein HY5 HOMOLOG (HYH) may interact with 12 members of the litchi LcBBX genes, and these 12 members (except LcBBX21) all belong to the Group IV and Group V subfamilies. GATA2 of the GATA transcription factor family, as a positive regulator of photomorphogenesis, only interacts with BBX genes of the Group IV subfamily (Figure 6). These findings provide an important reference for further analyzing the molecular regulatory network of BBX genes in litchi pericarp.

### 2.7. Prediction of miRNA Targets for Members of the LcBBX Genes

Predicting the miRNA targets of *LcBBX* family members is helpful for further studying the interaction between miRNAs and *LcBBXs*, as well as their functions in biological processes. The results showed that 16 litchi *BBX* genes have miRNA target binding sites (Appendix A). Among them: *LcBBX5/9/14/16* have miRNA target sites in both the CDS region and UTR region; *LcBBX6*/*7*/*8*/*11*/*13*/*15*/*17*/*18*/*19*/*20* have one or more miRNA target sites in the CDS region; *LcBBX10* has miRNA target sites only in the UTR region.

### 2.8. Tissue-Specific Expression of LcBBX Gene Family Members

To explore the potential functions of *LcBBX* genes and their roles in growth and development, this study analyzed the expression patterns of litchi *BBX* genes in different tissues and developmental processes. The results showed that the gene expression characteristics of litchi *LcBBX* exhibited obvious tissue specificity: *LcBBX17/19* were specifically highly expressed in male flowers, while all *LcBBX* genes maintained low transcriptional levels in female flowers; *LcBBX6/21/18/9/1/20/13/4/8* showed high transcriptional levels in leaves; In fruits, *LcBBX3/14/6/21/18/9/1/20/12/5* were highly expressed in the pericarp; *LcBBX10/8/5* had high transcriptional levels in the aril. Only *LcBBX11* was highly expressed in small fruits and seeds (Figure 7, Appendix A). These findings suggest that *LcBBX* may play a more important role in pericarp development and color formation in litchi.

To further reveal the potential functions of *LcBBX* in the process of litchi fruit development, this study analyzed the gene expression characteristics of *LcBBX* genes at different fruit development stages. The results showed that most *LcBBX* genes (*LcBBX15/14/3/9/2/13/7/1/16/20*) had high transcriptional levels before pericarp coloration, while *LcBBX5/8/12/17/18* had relatively high expression levels during the fruit coloration stage. At the fruit ripening stage, *LcBBX21/6/19/10/4* were highly expressed, followed by slightly lower expression of genes such as *LcBBX13/7/1/16*. The expression levels of *LcBBX4/10/19* gradually increased with pericarp development and ripening, which was consistent with the accumulation trend of anthocyanins in the pericarp, suggesting that they may be involved in the regulation of anthocyanin biosynthesis (Figure 8, Appendix A). The expression patterns of *LcBBX* genes exhibit distinct stage-specificity and complementary characteristics, providing important clues for investigating the functions of *LcBBX* genes during fruit developmental stages.

### 2.9. qRT-PCR Analysis of Expression Patterns of LcBBX Genes in Litchi Pericarp at Different Developmental Stages

qRT-PCR was used to further analyze the gene expression characteristics of litchi fruit at three stages: pre-coloration (50 DAFB), coloration (50 DAFB), and ripening (70 DAFB). The results showed that most *LcBBX* genes (*LcBBX1/2/3/9/14/15/16/21*) had high expression levels before pericarp coloration, decreased expression during the coloration stage, and then increased again at the ripening stage. This pattern was similar to the transcriptome data of different developmental stages (Figure 8). In contrast, the expression levels of *LcBBX4/5/6/13* increased during the pericarp coloration stage. In addition, *LcBBX10/11/12/17* were highly expressed during the pericarp coloration and ripening stages, while *LcBBX8* was highly expressed at the pericarp ripening stage (Figure 9). In summary, the expression levels of most *LcBBX* genes change during the developmental stages of litchi pericarp, suggesting that they may directly or indirectly participate in the anthocyanin accumulation process in litchi pericarp.

### 2.10. qRT-PCR Analysis of Expression Patterns of LcBBX Genes in Litchi Pericarp Under Light Induction

Previous studies have shown that light can induce the biosynthesis of anthocyanins in litchi pericarp [42]. The qRT-PCR results revealed that after de-shading of litchi fruits, the expression levels of multiple *LcBBX* genes changed significantly with light exposure time: the expression levels of *LcBBX1/7/15* increased on the first day after light recovery; *LcBBX4/5/6/8/19* were upregulated on the third day after light recovery. In contrast, the expression levels of *LcBBX3/9/12/14/18* decreased after light recovery (Figure 10). In conclusion, *LcBBX* genes exhibit diverse expression patterns in litchi pericarp during the light stage after de-shading. This differential expression characteristic suggests that they may play different regulatory roles in the light response of litchi pericarp and the subsequent anthocyanin accumulation, providing a basis for further analyzing the functions of *LcBBX* genes in processes such as light signal transduction and anthocyanin biosynthesis in litchi pericarp.

## 3. Discussion

At present, in addition to the MBW complex, transcription factors such as *NAC* [39], *bZIP* [40], and *ERF* [41] have been successively identified and confirmed to be involved in anthocyanin accumulation in litchi fruits. The *BBX* transcription factor plays a crucial role in regulating plant growth and development, and it may also play an important part in litchi fruit development and color formation. In this study, 21 *LcBBX* genes were screened from the litchi genome, and they were unevenly distributed on 12 chromosomes (Figure 3). In terms of gene number, *BBX* family members in litchi are fewer than in *Arabidopsis thaliana* (32) [3] and rice (30) [43], but higher than in pineapple (19) [44] and Chinese chestnut (18) [45], indicating that the distribution of *BBX* genes varies significantly among different plant species. Generally, gene duplication events such as segmental duplication and tandem duplication are the core driving forces for plant genome evolution and environmental adaptation [46]. The results of collinearity analysis showed that there were 4 pairs of segmentally duplicated genes and 1 pair of tandemly duplicated genes in the *LcBBX* gene family (Figure 3), which might be an important pathway for the expansion of the litchi *LcBBX* gene family. Collinearity analysis with other plant species revealed that the number of *BBX* homologous genes between litchi and other dicotyledonous plants was significantly higher than that between monocotyledonous plants. This suggests that dicotyledonous and monocotyledonous plants have undergone different gene duplication events during their evolutionary process, leading to the differentiation of *BBX* genes between the two types of plants (Figure 4). Phylogenetic tree analysis divided the 21 *LcBBX* family genes into 5 subfamilies, and members of the same subfamily had a consistent combination of conserved domains. This classification pattern is highly consistent with that in *Arabidopsis* and rice, implying the evolutionary conservation of the *BBX* gene family classification across different plants (Figure 1). Meanwhile, the three types of conserved domains (B-box1, B-box2, and CCT) contained in *LcBBX* genes have highly similar sequences, which further confirms the stability of the conserved domains of the *LcBBX* family during evolution and provides a structural basis for the subsequent analysis of their functional conservation (Figure 2 and Appendix A).

*Cis*-acting elements in the promoter region play a crucial regulatory role in the functional diversity of genes [47], and they are mainly classified into four categories: light-responsive elements, hormone-responsive elements, stress-responsive elements, and growth and development elements. Among the *cis*-acting elements in the promoters of *LcBBX* genes, the light-responsive element G-box is the most abundant (Figure 5). Previous studies have confirmed that the transcription factor *HY5* can regulate the expression of downstream genes by binding to the G-box in their promoter regions [48], and BBX proteins can serve as rate-limiting cofactors for *HY5* [49]. In *Arabidopsis thaliana*, *HY5* can interact with *AtBBX21/22/23* to collectively regulate plant photomorphogenesis [23,24,25]. Meanwhile, GO functional enrichment analysis showed that most *LcBBX* family members are significantly enriched in light response-related terms, suggesting that they may be involved in the light signal response process of litchi (Appendix A). Among the hormone-responsive elements, the ABA-responsive element ABRE is the most numerous and widely distributed in the promoters of *LcBBX* genes (Figure 5 and Appendix A). Additionally, existing studies have verified that ABA can promote anthocyanin accumulation in litchi pericarp [50]. These findings indicate that *LcBBX* genes may respond to both light and ABA signals, and participate in the integration of light and ABA signal pathways by regulating the expression of downstream structural genes, thereby modulating the biosynthesis of anthocyanins in litchi fruits. The results of protein–protein interaction prediction revealed that there are numerous interaction relationships among members of the *LcBBX* gene family, and they may exert regulatory functions synergistically by forming protein complexes; LcBBX proteins can also interact with light-responsive factors such as HY5, HYH, and GATA2 (Figure 6). It can be concluded that *LcBBX* genes play important roles in responding to light and ABA signals and regulating anthocyanin biosynthesis.

*BBX* genes are involved in regulating anthocyanin biosynthesis in various plant species, and their functions are often associated with light response [20,51,52]. In this study, the expression patterns of *LcBBX* family members exhibited distinct tissue-specificity, among which 10 *LcBBX* genes (*LcBBX1/3/5/6/9/12/14/18/20/21*) showed high expression levels in the pericarp (Figure 7). Most *LcBBX* genes were highly expressed before pericarp coloration. Furthermore, the expression levels of *LcBBX4/10/19* gradually increased with the development of the pericarp, which was almost consistent with the accumulation trend of anthocyanins in the pericarp (Figure 8). Similarly, in *Rubus chingii*, *RcBBX* genes also displayed differential expression patterns in roots, stems, leaves, flowers, and fruits, and most *RcBBX* genes were highly expressed before pericarp coloration during fruit development [53]. This shows a certain similarity to the expression pattern of *LcBBX* genes in litchi, suggesting that the function of *BBX* genes in regulating fruit development and anthocyanin synthesis processes has a certain evolutionary conservation. qRT-PCR results showed that as the coloration process of litchi pericarp progressed, the expression levels of genes such as *LcBBX4/5/6/10* increased significantly (Figure 9). Meanwhile, multiple *LcBBX* genes changed obviously after the shading treatment was removed: the expression levels of *LcBBX1/4/5/6/7/8/15/19* increased significantly within 1–3 days after light recovery, while the expression levels of *LcBBX9/12/14/18* decreased significantly after light recovery (Figure 10). It can be seen that *LcBBX* genes have obvious light-responsive characteristics and may play a key regulatory role in the light-induced anthocyanin biosynthesis process in litchi pericarp.

## 4. Materials and Methods

### 4.1. Plant Materials and Sources of Public Data

The ‘Feizixiao’ litchi cultivar was cultivated at the Litchi and Longan Experimental Base, Institute of South Subtropical Crops, Chinese Academy of Tropical Agricultural Sciences. For the purpose of litchi pericarp sampling at various developmental stages, samples were collected at 50 days after full bloom (DAFB, pre-coloration stage), 60 DAFB (coloration stage), and 70 DAFB (ripening stage). At each developmental stage, three trees were selected as biological replicates, and 10 fruits were uniformly collected from different orientations of each tree. The pericarps were then peeled, immediately flash-frozen in liquid nitrogen, and preserved. For the sampling of litchi fruits under shading treatment, at 42 DAFB, three litchi trees exhibiting consistent growth vigor and normal fruit set were randomly selected, and 10 branches per tree were subjected to shading treatment. The shading treatment concluded at 63 DAFB, and pericarp samples were collected at 0 days after shading removal, as well as at 1, 3, and 7 days post-unshading. At each time point, 10 shaded fruits were randomly harvested from each of the three trees, and the fruits collected from each individual tree were stored appropriately.

In addition, transcriptome data of different litchi tissues were obtained from the accession number PRJNA747875 (https://www.ncbi.nlm.nih.gov/bioproject/PRJNA747875 (accessed on 4 March 2024)), including pericarp, aril (flesh), seed, fruitlet, carpopodium, male flower, female flower, stigma, stamens, leaf, sterile stamens, and ovary. Transcriptome data of the litchi fruit coloration process were derived from the accession number PRJNA261000 (https://www.ncbi.nlm.nih.gov/bioproject/PRJNA261000 (accessed on 8 March 2024)), covering the green (pre-coloration), yellow (coloration stage), and red (ripening stage).

### 4.2. Identification of LcBBX Gene Family Members

Litchi genome information, including DNA, GFF, CDS, and protein files, was downloaded from SapBase (http://www.sapindaceae.com/Download.html (accessed on 17 October 2024)). *Arabidopsis thaliana* genome data were downloaded from TAIR (https://www.arabidopsis.org/ (accessed on 17 October 2024)). Protein sequences of *AtBBX* family members were downloaded from the PlantTFDB database (https://planttfdb.gao-lab.org/ (accessed on 17 October 2024)) as query sequences, and the BLAST tool in TBtools software (version 2.363) was used to search for homologous sequences in the litchi protein database. In addition, the hidden Markov model (HMM) of the BBX zinc finger domain (PF00643) was downloaded from the InterPro database (https://www.ebi.ac.uk/interpro/entry/pfam/PF00643/ (accessed on 21 October 2024)), and the HMMER software package (version 3.0) was used for homologous sequence analysis (threshold of 1 × 10^−5^). Using AtBBX proteins as query sequences, TBtools v2.363 was then employed to search for putative LcBBX proteins (e value < 1 × 10^−5^ and coverage ≥ 50%). The search results of HMMER and BLAST were combined as candidate members of the litchi *BBX* family. The conserved domains in protein sequences were analyzed using the Web CD-search tool of NCBI (https://www.ncbi.nlm.nih.gov/Structure/cdd/wrpsb.cgi (accessed on 23 October 2024)), and protein sequences without the B-box conserved domain were excluded to finally obtain the members of the litchi *BBX* gene family. The ExPASy website (https://web.expasy.org/protparam/ (accessed on 27 October 2024)) was used to analyze parameters such as relative molecular weight, number of amino acids, isoelectric point, and instability index. The WoLF PSORT website (https://wolfpsort.hgc.jp/ (accessed on 27 October 2024)) was used to predict subcellular localization.

### 4.3. Construction of the Phylogenetic Tree of LcBBX Genes

MEGA11 software (version 11.0.13) was used to construct the phylogenetic tree of the *BBX* gene family members from litchi, *Arabidopsis*, rice, pear, and pineapple. ClustalW was used for multiple sequence alignment of all protein members. The neighbor-joining (NJ) method was adopted, with 1000 bootstrap replicates for reliability testing; other parameters were set to default, and the phylogenetic tree was plotted.

### 4.4. Analysis of Conserved Domains, Gene Structure, and Conserved Motifs of LcBBX Genes

Protein sequences of litchi *BBX* genes were submitted to the CD-search Tool (https://www.ncbi.nlm.nih.gov/Structure/cdd/wrpsb.cgi (accessed on 23 October 2024)) for conserved domain prediction, and were visualized using the Gene Structure View tool in TBtools software (version 2.363). Using the litchi genome GFF file, the Gene Structure View tool in TBtools software was used to analyze and visualize the gene structure of litchi *BBX* genes. The MEME website (https://meme-suite.org/meme/ (accessed on 3 December 2024)) was used to analyze protein conserved motifs, with the number of motifs set to 10 (other parameters are default values), and were visualized using the Gene Structure View tool in TBtools software.

### 4.5. Chromosomal Localization and Collinearity Analysis of LcBBX Genes

Gene files and genome annotation files of litchi, as well as those of longan and rambutan (*Sapindaceae*), *Arabidopsis* and apple (dicotyledons), and pineapple and rice (monocotyledons), were used. The One Step MCScanX tool in TBtools software was employed to visualize intraspecific collinearity of litchi and interspecific collinearity between litchi and other species, so as to analyze the evolutionary relationship of *BBX* genes. Data of longan and rambutan were obtained from SapBase (http://www.sapindaceae.com/Download.html (accessed on 6 August 2024)); data of *Arabidopsis* were from the TAIR database (https://www.arabidopsis.org/ (accessed on 17 October 2024)); data of rice and apple were from the Ensembl database (http://plants.ensembl.org/index.html (accessed on 13 March 2025)); and pineapple data were from the Ananas Genomic Database (http://pineapple.zhangjisenlab.cn/pineapple/html/download.html (accessed on 13 March 2025)).

### 4.6. Analysis of cis-Acting Elements in LcBBX Gene Promoters

The Gtf/Gff3 Sequences Extract tool in TBtools software v2.363 was used to extract the 2000 bp gene sequence upstream of the start codon of *LcBBX* genes as the promoter region. Promoter sequences were submitted to the Plant CARE website (http://bioinformatics.psb.ugent.be/webtools/plantcare/html/ (accessed on 12 March 2025)) for *cis*-acting element prediction, and the results were visualized using the Simple BioSequence Viewer tool in TBtools software v2.363.

### 4.7. GO Enrichment and Protein–Protein Interaction Prediction of LcBBX Genes

Protein sequences of *LcBBX* genes were submitted to the online STRING website (https://cn.string-db.org/) for protein–protein interaction prediction of litchi BBX proteins. The Functional Enrichment Visualization tool on this website was used to perform GO enrichment analysis of *LcBBX* genes.

### 4.8. Prediction of miRNA Targets of LcBBX Genes

Litchi miRNA annotation files were downloaded from SapBase (http://www.sapindaceae.com/sRNA/litchi-miRNA-list.html (accessed on 14 March 2025)). The psRNATarget website (https://www.zhaolab.org/psRNATarget/ (accessed on 14 March 2025)) was used to predict microRNA (miRNA) target binding sites in the coding sequence (CDS) and untranslated region (UTR) of *LcBBX* genes, with the expected value set to ≤4.5 and other parameters set to default.

### 4.9. Quantitative Real-Time PCR (qRT-PCR) Analysis of LcBBX Genes at Different Pericarp Developmental Stages

Total RNA was isolated utilizing the Fast Universal Plant RNA Extraction Kit (Huayueyang, Beijing, China). The purity of the RNA, indicated by OD_260/280_ and OD_260/230_ ratios, was assessed with a NanoDrop UV-Vis spectrophotometer (Thermo Fisher Scientific, Waltham, MA, USA), and RNA integrity was examined using an Agilent 2100 Bioanalyzer (Agilent Technologies, Santa Clara, USA). Reverse transcription of the total RNA to synthesize cDNA for quantitative real-time PCR (qRT-PCR) analysis was conducted using the RevertAid First Strand cDNA Synthesis Kit (Thermo Fisher Scientific, Waltham, MA, USA). qRT-PCR primers were designed with the Batch q-PCR Primer Design plugin in TBtools software version 2.363 (Appendix A). The *MaActin* gene (GenBank accession number: HQ615689) served as the internal reference [50]. qRT-PCR was executed on a QuantStudio 1 Real-Time PCR system following the manufacturer’s protocol for the ChamQ Universal SYBR qPCR Master Mix (Vazyme, Nanjing, China). The thermal cycling conditions were as follows: initial denaturation at 95 °C for 30 s, followed by 40 cycles of 95 °C for 10 s and 60 °C for 30 s. Each reaction was performed in three technical replicates. The relative expression levels of *LcBBXs* were calculated using the 2^−∆∆Ct^ method [54]. Data visualization was carried out using Origin 2024 software.

### 4.10. Statistical Analysis

The qRT-PCR data underwent a one-way analysis of variance (ANOVA) using SPSS (version 23.0), followed by Tukey’s multiple comparison test to conduct statistical analyses. Prior to performing the ANOVA tests, the assumptions of variance homogeneity and normality were meticulously verified to ensure the validity of the statistical conclusions [55]. Each experiment comprised three independent biological replicates, and the qRT-PCR results are reported as the mean ± standard error (SE) of these replicates. Statistical significance was determined at a threshold of *p* < 0.05, with different letters indicating significant differences among the samples.

## 5. Conclusions

In this study, 21 *LcBBX* genes (*LcBBX1*-*LcBBX21*) were identified from the litchi genome, which are unevenly distributed across 12 chromosomes. Based on phylogenetic relationships, these *LcBBX* members can be divided into 5 subfamilies (Group I to V). The promoter regions of *LcBBX* genes are rich in the light-responsive element G-box and the ABA-responsive element ABRE. Protein interaction prediction revealed that most LcBBX proteins can interact with HY5, suggesting that *LcBBX* may be involved in the regulation of physiological processes by integrating light and ABA signals. In terms of gene expression characteristics, *LcBBX* genes exhibit distinct tissue-specific expression patterns. Most members are highly expressed before pericarp coloration; in addition, *LcBBX4/10* are upregulated during the fruit coloration stage. After the removal of shading treatment, the expressions of *LcBBX1/4/6/7/15/19* are upregulated in response to light. In conclusion, *LcBBX4* may directly participate in the regulation of anthocyanin biosynthesis in litchi pericarp. This study clarifies the basic characteristics and functional potential of the *LcBBX* family in litchi, and provides an important reference for subsequent functional verification of *LcBBX* genes and research on the transcriptional regulatory mechanism of litchi anthocyanin biosynthesis.

## Figures and Tables

**Figure 1 ijms-26-10834-f001:**
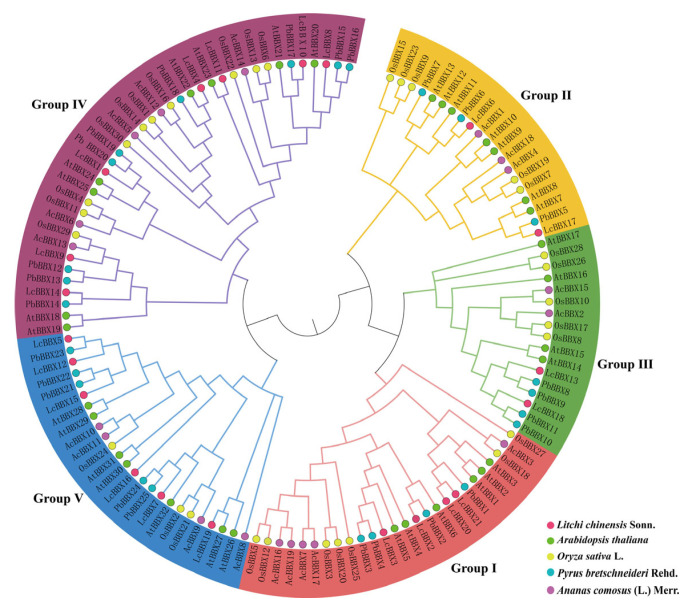
Phylogenetic relationships of the *BBX* gene family in litchi, *Arabidopsis*, rice, pear, and pineapple. Different coloured branches represent different subfamilies. The red, green, yellow, blue, and purple circular shapes represent the *BBX* genes of litchi, *Arabidopsis*, rice, pear, and pineapple.

**Figure 2 ijms-26-10834-f002:**
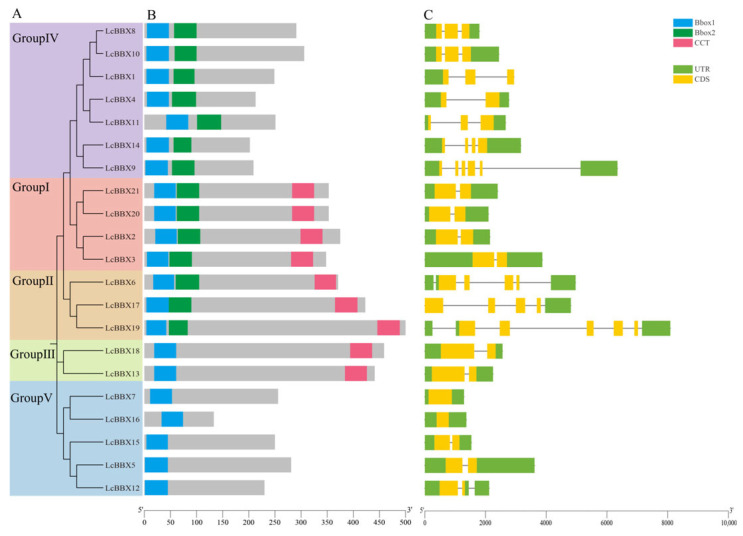
Characterization of *LcBBXs*. (**A**) neighbor-joining tree analysis and its grouping of *LcBBXs*; (**B**) conserved structural domains of *LcBBXs*. The three conserved structural domains are represented by boxes of three colors; (**C**) gene structure of *LcBBXs*. Green and yellow boxes indicate UTR and CDS.

**Figure 3 ijms-26-10834-f003:**
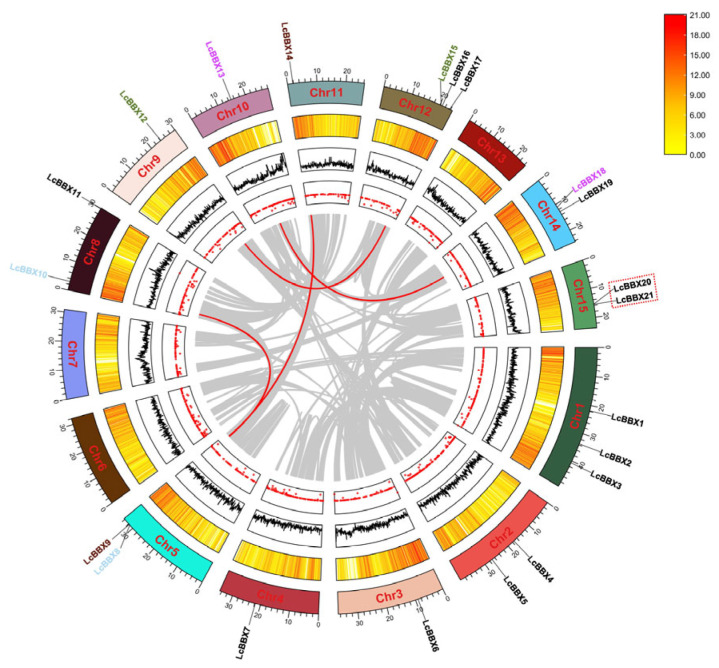
Intraspecific collinearity analysis of the *LcBBX* gene family members. The segmental duplicated gene pairs are linked by red lines, and the tandem duplicated gene pairs are marked with red dashed lines. The figure from outside to inside shows gene chromosome localization, gene density, genomic GC ratio, and genomic gap statistics.

**Figure 4 ijms-26-10834-f004:**
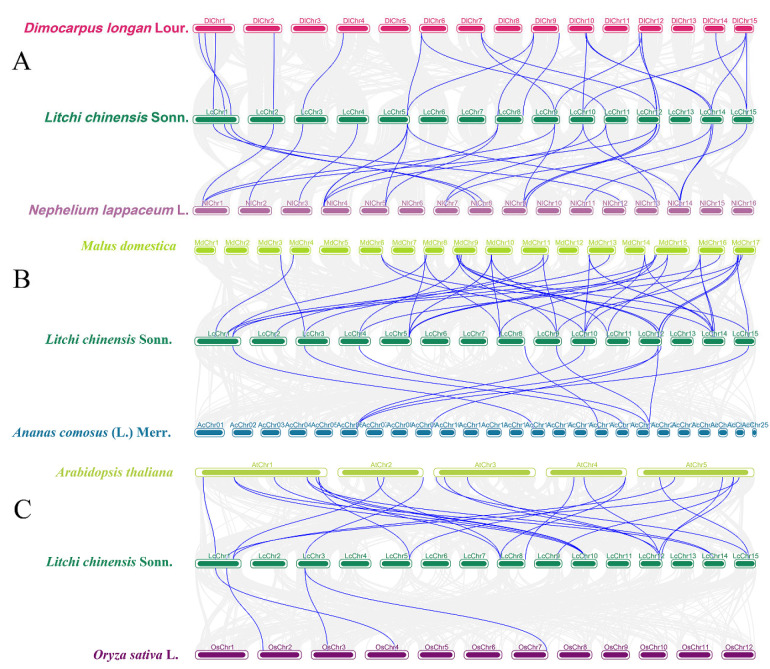
Collinearity analysis of the BBX gene family across multiple species. (**A**) Collinearity analysis between litchi, longan, and rambutan. (**B**) Collinearity analysis between litchi, apple, and pineapple. (**C**) Collinearity analysis between litchi, *Arabidopsis,* and rice. Gray lines represent the collinear within litchi and other plant genomes, blue lines emphasize the pairs of syntenic *BBX* genes.

**Figure 5 ijms-26-10834-f005:**
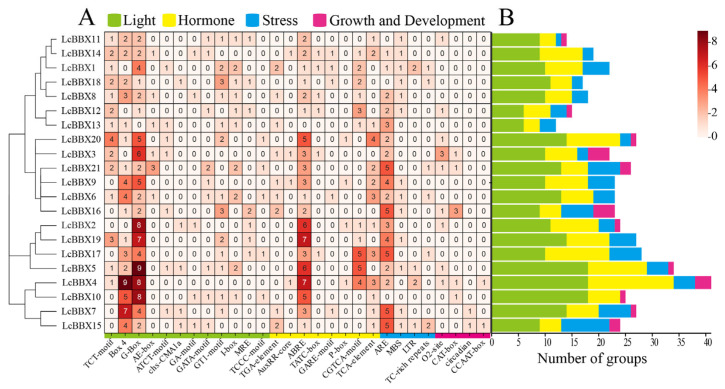
Statistics of *cis*-acting elements of promoters in *LcBBX* genes. The green, yellow, blue, and pink modules represent light-responsive elements, hormone-responsive elements, stress-responsive elements, and growth and development elements. (**A**) number of each element in *LcBBXs*; (**B**) number of elements per group in *LcBBXs*.

**Figure 6 ijms-26-10834-f006:**
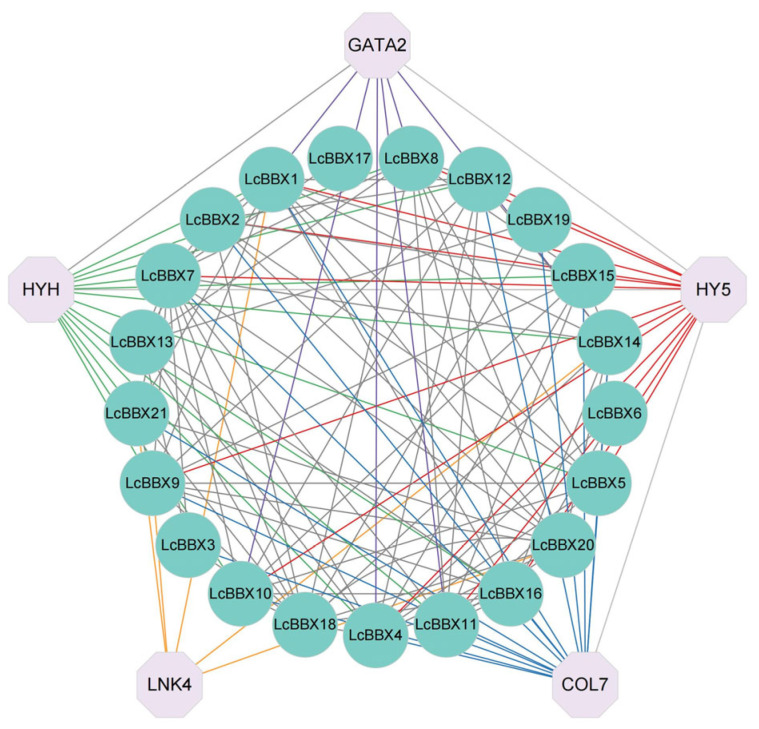
Protein–protein interaction network of *LcBBX* proteins. The circle represents *LcBBXs*, and the octagon represents genes that interact with *LcBBXs*. Each gene that interacts with *LcBBX* is connected using lines of different colors. The red lines indicate LcBBXs that interact with HY5; the green lines indicate that LcBBXs interact with HYH; the purple lines indicate that LcBBXs interact with GATA2.

**Figure 7 ijms-26-10834-f007:**
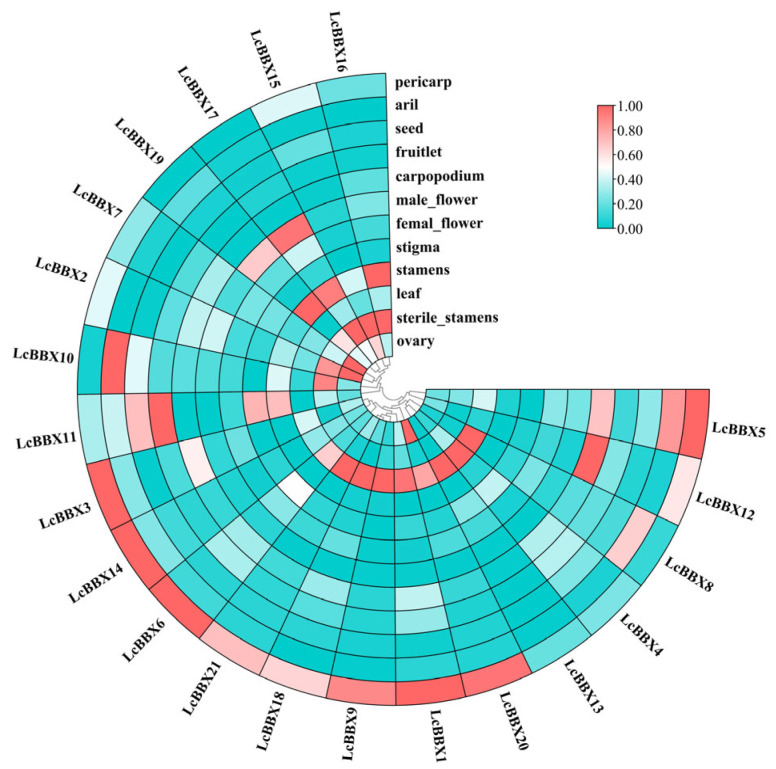
The expression profiles of litchi *BBX* genes in different tissues. A heatmap shows the relative expression levels in pericarp, aril (flesh), seed, fruitlet, carpopodium, male flower, female flower, stigma, stamens, leaf, sterile stamens, and ovary.

**Figure 8 ijms-26-10834-f008:**
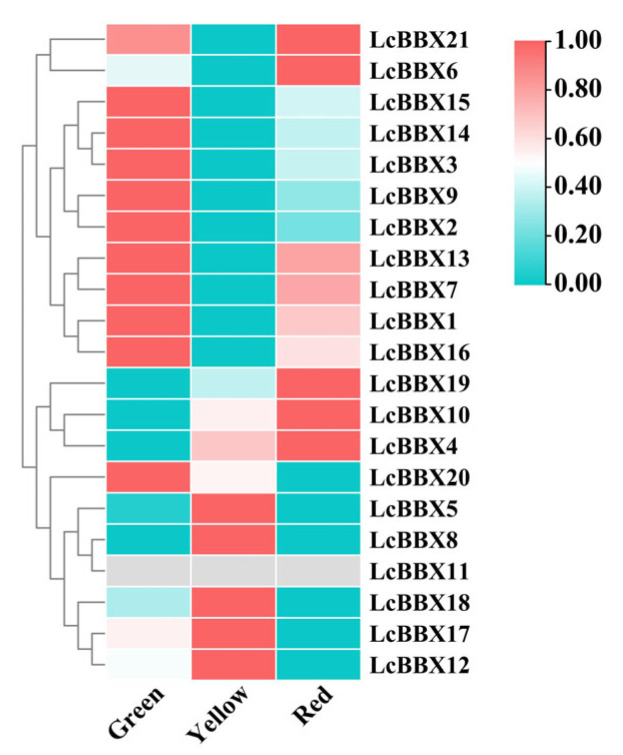
The expression profiles of *LcBBX* genes at different stages of pericarp coloring. The three developmental stages include green pericarp, yellow pericarp, and red pericarp.

**Figure 9 ijms-26-10834-f009:**
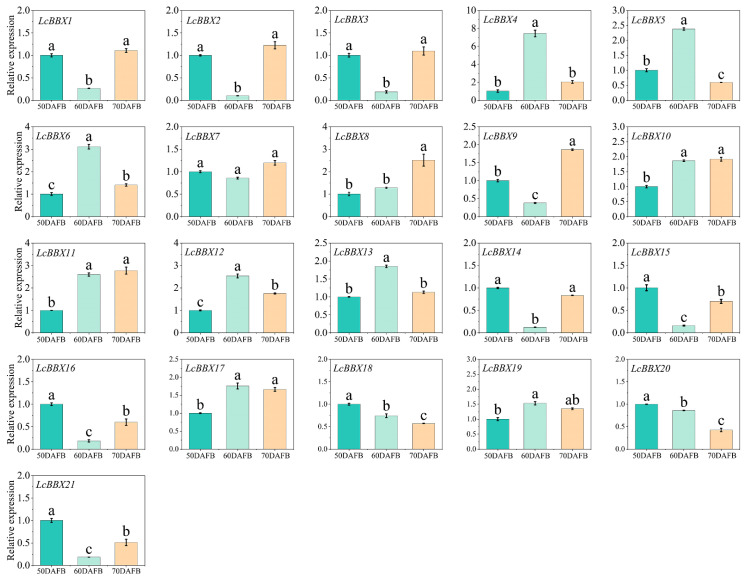
Expression patterns of *LcBBX* genes in different stages of pericarp development, namely the pre-coloration stage (50 DAFB), coloration stage (60 DAFB), and ripening stage (70 DAFB) of litchi pericarp. Data are means  ±  SEM of three biological replicates. Different letters mean significant differences at the 0.05 level.

**Figure 10 ijms-26-10834-f010:**
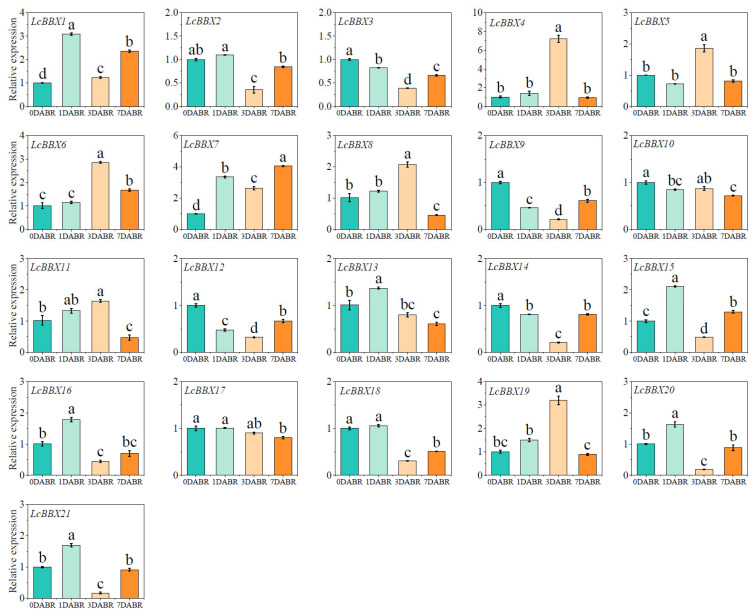
Expression patterns of *LcBBX* genes in litchi pericarp with the removal of shading treatment. DABR, days after bag removal. Data are means  ±  SEM of three biological replicates. Different letters mean significant differences at the 0.05 level.

## Data Availability

The original contributions presented in the study are included in the article and Appendix A, further inquiries can be directed to the corresponding authors.

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
