# Peer review of "Genome-Wide Identification of the Litchi BBX Gene Family and Analysis of Its Potential Role in Pericarp Coloring"

_ijms, 2025, doi:10.3390/ijms262210834_

Round 1
Reviewer 1 Report
Comments and Suggestions for Authors
This study includes the genome-wide analysis of BBX family proteins in the litchi fruit species. And suspect that these genes might be important for light-induced anthocyanin regulation in the fruit. Most of the results are based on prediction-based analysis. While supplemented with previously published RNA-seq data, qPCR analysis was conducted on the basis of RNA-seq observation. However, no confirmational experiments were conducted. It is suggested to supplement functional analysis to have a stronger conclusion.
Other comments:
In Figure 1, for phylogenetic analysis, why did the author not choose other species, including fruits? Kindly add other species to better understand the phylogenetic relationship of the BBX family among various species.
In Figure 4, for synteny analysis, it is understood that the longan and rambutan belong to the same family. While for important fruit species, why did the author choose grape and pineapple (why not mango, apple, citrus, pear, peach, or other fruit species), and for model plant species, why did the author choose the tomato and rice and not Arabidopsis, as there has been intensive work done in Arabidopsis?
In Figure 2D, the font is not visible.
Why was the only influence of light seen in litchi fruit? While ABA responsive element (ABRE) and drought-induced response element (MBS) also have major cis-element sites in the promoter region of BBX family members. Then why was ABA or drought treatment not applied? Kindly supplement data or explain.
Why is protein interaction of the BBX family observed in only 5 proteins? Kindly explain
In Figures 8 and 9, the relative expression of most of the BXX genes does not show a similar trend. Kindly explain.
What does DABR represent in figure 10? Kindly also add in the figure legend. Moreover, in 0, 1, 3, and 7 days no significant trend was seen among most of the gene expression; is it reasonable to conclude that BBX genes might play a role in light-induced anthocyanin biosynthesis? Kindly explain.
The conclusion is not strong, as the author suggested that the LcBBX4 might be important for anthocyanin regulation, but in figures 10 and 11, the expression is only higher at the yellow stage and 3DABR, respectively. In my opinion, on this basis, it is not suitable to conclude a gene. It is suggested to perform functional confirmation and supplement the manuscript with at least overexpression or silencing analysis of the putative gene to support the conclusion.
Mistakes were also found in the manuscript, such as species names (lines 340, 377), and the use of ‘cis’ should be italicized.
Mistake in line 276.
Author Response
Response to Reviewer 1
Thank you very much for your valuable comments. According to your comments, we have revised the shortcomings in the manuscript one by one.
- Comment: While supplemented with previously published RNA-seq data, qPCR analysis was conducted on the basis of RNA-seq observation. However, no confirmational experiments were conducted. It is suggested to supplement functional analysis to have a stronger conclusion.
Response: We sincerely appreciate the valuable suggestions you have proposed. We fully agree that directly verifying the function of BBX genes through functional experiments and mechanism elucidation would provide more robust support for the conclusions. However, the primary objective of this study is to conduct the first genome-wide systematic identification and analysis of the litchi BBX gene family. By integrating phylogenetic analysis, identification of conserved domains, analysis of cis-acting elements in promoters, and combined with independent qRT-PCR validation of its expression pattern associated with light-induced anthocyanin accumulation, this study can provide entry points and insights for subsequent functional research.
Meanwhile, the regulatory role of BBX transcription factors in modulating litchi anthocyanin accumulation by responding to light and plant hormone signals, as well as the associated functional validation (both heterologous and homologous), are also under systematic investigation. Nevertheless, research on functional genes in woody fruit trees is a technically complex and time-consuming process, and the results of subsequent studies will be further presented once they are refined. We believe that as a solid genomic resource paper, this study has achieved sufficient comprehensiveness in its data and conclusions, and provides crucial value to this research field. We would like to express our gratitude again for your insightful comments..
- Comment: In Figure 1, for phylogenetic analysis, why did the author not choose other species, including fruits? Kindly add other species to better understand the phylogenetic relationship of the BBX family among various species.
Response: According to your suggestion, we have added the BBX genes of pear and pineapple in Figure 1 for the phylogenetic analysis and revised the relevant descriptions.
- Comment: In Figure 4, for synteny analysis, it is understood that the longan and rambutan belong to the same family. While for important fruit species, why did the author choose grape and pineapple (why not mango, apple, citrus, pear, peach, or other fruit species), and for model plant species, why did the author choose the tomato and rice and not Arabidopsis, as there has been intensive work done in Arabidopsis?
Response: Based on your suggestion, in Figure 4, we have added the BBX gene sequences of Arabidopsis and apple for collinearity analysis and revised the relevant descriptions. Arabidopsis is a classic model plant, and its BBX genes have been extensively studied; apple is a typical woody fruit tree, and the functional research of its BBX genes has also received considerable attention.
- Comment:In Figure 2D, the font is not visible.
Response: We have modified the layout and clarity of Figure 2.
- Comment:Why was the only influence of light seen in litchi fruit? While ABA responsive element (ABRE) and drought-induced response element (MBS) also have major cis-element sites in the promoter region of BBX family members. Then why was ABA or drought treatment not applied? Kindly supplement data or explain.
Response: This is a very good suggestion. The BBX gene of plants is an important transcription factor in the light-responsive pathway. It mainly functions by interacting with HY5 in response to light signals. Therefore, our research focused on the light-responsive cis-acting elements. Abscisic acid (ABA) is also an important influencing factor in the regulation of anthocyanins. In the subsequent studies, we will also further pay attention to the synergistic regulatory effect of light signals and ABA signals. Unfortunately, because we currently do not have the ability to treat lychee fruits with abscisic acid and its inhibitors, we are temporarily unable to conduct the relevant experiments and cannot provide the necessary supplementary data.
- Comment:Why is protein interaction of the BBX family observed in only 5 proteins? Kindly explain.
Response: In fact, we conducted interaction protein prediction for all members of the BBX protein family. Since our focus was on the BBX proteins that regulate light signals, we selected five proteins that respond to light and are involved in photomorphogenesis for mapping analysis.
- Comment: In Figures 8 and 9, the relative expression of most of the BXX genes does not show a similar trend. Kindly explain.
Response: Except for LcBBX6/4/13, the overall trends of the other members are similar. For example, the expression levels of members such as LcBBX1/2/3/14/15/16/21 all decreased first and then increased. The reason for this partial inconsistency or slight differences in trends might be that the transcriptome data comes from literature, while the samples for quantitative analysis were collected by our laboratory. There might be some differences in the sampling and processing times between the two, which could lead to slight differences in expression patterns.
- Comment: What does DABR represent in figure 10? Kindly also add in the figure legend. Moreover, in 0, 1, 3, and 7 days no significant trend was seen among most of the gene expression; is it reasonable to conclude that BBX genes might play a role in light-induced anthocyanin biosynthesis? Kindly explain.
Response: We have provided the full name of DABR as "Days After Bag Removal".
The expression levels of LcBBX1/7/15 increased on the first day after light recovery; LcBBX4/5/6/8/19 were upregulated on the third day after light recovery. The expression of these genes is induced by light signals, and shows a strong consistency with the coloring of the fruit peel and the accumulation of anthocyanins. Therefore, it can be inferred that "the BBX gene may play a role in light-induced anthocyanin biosynthesis".
- Comment:The conclusion is not strong, as the author suggested that the LcBBX4 might be important for anthocyanin regulation, but in figures 10 and 11, the expression is only higher at the yellow stage and 3DABR, respectively. In my opinion, on this basis, it is not suitable to conclude a gene. It is suggested to perform functional confirmation and supplement the manuscript with at least overexpression or silencing analysis of the putative gene to support the conclusion.
Response: LcBBX4 is highly expressed during the color-changing period (yellow fruit stage) of fruit development; 3DABR is the period when the anthocyanin content in the fruit peel of litchi rapidly accumulates after removing shading. The expression level of LcBBX4 is also significantly higher than at other time points. There are two pieces of evidence that suggest that the LcBBX4 gene plays an important role in the regulation of anthocyanin synthesis. Indeed, your suggestion is also very valuable. To prove that LcBBX4 regulates the synthesis of anthocyanins in litchi, experiments such as transient and stable genetic transformation are needed to confirm it. Besides functional verification, systematic research on the molecular mechanism of LcBBX4 transcriptional regulation also needs to be carried out.
- Comment: Mistakes were also found in the manuscript, such as species names (lines 340, 377), and the use of ‘cis’ should be italicized.
Response: Thank you very much for your suggestion. We have already corrected these unnecessary mistakes.
- Comment: Mistake in line 276.
Response: We have already corrected this mistake.

Reviewer 2 Report
Comments and Suggestions for Authors Authors analysed the B-box (BBX) transcription factors family in Litchi (Litchi chinensis Sonn.). Authors provided the basic physicochemical properties of BBX proteins. Also the chromosomal localization of BBX genes as well as BBX proteins domain organization was presented. Basic evolutionary informations as dupliaction events were presented, together with the collinearity relationships between Litchi, related species and model plants as Arabidopsis, or rice. Ditribution of cis-active elements within promoter regions of BBX genes provided informations related to the responsiveness to light, phytohormone and stress-conditions. Also miRNA targets among BBX genes in Litchi were identified, together with putative protein-protein interactions network. Results of available transcriptomic studies were used to characterize the tissue-specific and fruit development-specific gene expression. RT-PCR studies were used to evaluate the expression of BBX genes at different stages of pericarp development. Moreover, effects of light induction on BBX gene expression in Litchi pericarp was assessed. Study provides valuable results related to the role of BBX trans-factors in Litchi fruit development. Obtained results support conclusions, article is well written. Following comments should be addressed to improve the manuscript before the publication: 1. Fig. 9. Although this information is present in Materials and Methods, add to the description of Fig. 9 that Green means 50 days after full bloom (DAFB), similarly for Yellow - 60 DAFB and Red – 70 DAFB. 2. Section 4.9 Add following informations: I. How the purity and quality of RNA was assessed? II. How the putative remnants of genomic DNA were removed? III. Add citation of Livak and Schmittgen method. IV. If the stability of reference gene was not tested, (BestKeeper or related software) add the citation of preovious use of reference gene in RT-PCR. Other comments: Italicize the gene name in lines: 30,66,90,249,269,280,285,291,304. Line 317- do not italicize the name of protein. Line 80, 448- italicize Sapindaceae Line 252- do not capitalize O in the word Only Italicize Arabidopsis in lines 337,415,448,453. Adjust the references to the mdpi format: name of journal in italic, year in bold, volume in italic.Author Response
Response to Reviewer 2
Thank you very much for your valuable comments. According to your comments, we have revised the shortcomings in the manuscript one by one.
- Comment: Fig. 9. Although this information is present in Materials and Methods, add to the description of Fig. 9 that Green means 50 days after full bloom (DAFB), similarly for Yellow - 60 DAFB and Red – 70 DAFB.
Response: According to your suggestion, we have supplemented the relevant explanations in the description of Figure 9.
- Comment: Section 4.9 Add following informations: I. How the purity and quality of RNA was assessed? II. How the putative remnants of genomic DNA were removed? III. Add citation of Livak and Schmittgen method. IV. If the stability of reference gene was not tested, (BestKeeper or related software) add the citation of preovious use of reference gene in RT-PCR.
Response: According to your suggestion, we we have supplemented the detection of RNA purity and quality in the Materials and Methods section, and added descriptions of genomic DNA removal. Meanwhile, we have included citations for the Livak and Schmittgen method as well as for the reference genes.
- Comment: Other comments: Italicize the gene name in lines: 30,66,90,249,269,280,285,291,304. Line 317- do not italicize the name of protein. Line 80, 448- italicize Sapindaceae Line 252- do not capitalize O in the word Only Italicize Arabidopsis in lines 337,415,448,453.
Response: Thank you very much for pointing out some formatting errors in the writing of our paper. We have carefully revised these errors.
- Comment: Adjust the references to the mdpi format: name of journal in italic, year in bold, volume in italic.
Response: According to your suggestion, we have revised the format of all references in accordance with the requirements of MDPI.

Reviewer 3 Report
Comments and Suggestions for Authors
The authors identified 21 genes of the BBX family in the lychee genome, classified them (into 5 groups), analyzed domains/motifs, chromosomal location, and collinearity, analyzed promoter elements, predicted PPI and miRNA targets, and presented transcriptomic and qRT-PCR data for different tissues/stages and after shading removal. Based on this, they propose LcBBX4 (and partially LcBBX10/19) as a candidate involved in the regulation of anthocyanin skin coloration.
Advantages
1. Comprehensive “omics” approach: combining genomic search, structural/phylogenetic analysis, promoters, PPI, and expression (transcriptomes + qRT-PCR) provides a broad picture of the LcBBX family.
2. The experimental part with field shading and measurement of the response after removing the bags is useful for studying light-induced regulation.
3. There are practical conclusions and candidates for further verification.
Disadvantages
1. There is no experimental evidence of protein–protein or protein–DNA interactions (all PPI is predictive, based on STRING) — no co-IP/ Y2H/LC-MS/MS/EMSA/ChIP or transient transactional analyses to confirm LcBBX↔HY5 interaction or regulation of the LcMYB1 promoter, etc.
2. Lack of information on methodological parameters: for HMM/BLAST searches, the thresholds used (e-value, coverage) are not specified; for MEME, the e-value and search parameters are not specified; for phylogeny, only NJ is used—there is no maximum likelihood (ML) or Bayesian estimates and no indication of the evolutionary model.
3. No verification of protein localization/stability — predictions of subcellular localization (WoLFPSORT) and instability indices are not supported by experiments (GFP construct, Western blot).
4. qRT-PCR — important methodological data not specified: no information was found on reference genes for normalization, primer efficiency testing (standard curve), noise/melt curves, or the number of independent biological replicates (the methods specify 3 trees × 10 fruits, but it is unclear how this is aggregated for statistics). This weakens confidence in the quantitative analysis of expression.
5. Statistics: ANOVA + Tukey are used, but there is no mention of testing for normality/homogeneity of variances, multiple testing when analyzing a large set of genes (FDR, etc.). For transcriptomic data, it is also unclear whether an adjusted p-value was used.
6. The proposal that LcBBX4 is a “direct” regulator of anthocyanins is premature without functional validation (overexpression/RNAi/CRISPR; promoter-luciferase; EMSA/ChIP). The authors themselves provide indirect evidence (expression, promoter elements, PPI prediction), but this is not equivalent to functional confirmation.
Line-by-line notes:
28-29 Phrase "directly regulate anthocyanin biosynthesis" is unproven since the text only outlines a circle of candidates for regulators. I believe that the abstract should contain this understanding of the situation!
101-111 Here and in the corresponding methods section, BLAST/HMMER parameters (e-value, coverage, threshold) are not specified, and multiple domain alignments to e-values/coverage are not provided. Recommendation: add additional files (alignments, HMM scores) and selection criteria.
103-109 The stability prediction (instability index) and subcellular localization are useful, but experimental verification (GFP fusion) is necessary for key candidates (LcBBX4/10/19).
175-190 There is no data on the presence of conserved G-boxes in the promoters of key structural genes of the anthocyanin pathway (e.g., LcMYB1 or diosgenin genes), and there is no indicative promoter-luciferase analysis for LcBBX4.
210-236 PPI constructed via STRING—a predictive network—shows HY5/HYH connections. Criticism: STRING is useful as a hint, but does not replace experimental co-IP/BiFC/Y2H. Recommendation: conduct at least one experimental confirmation of the key interaction between LcBBX4 and HY5 (e.g., BiFC in Nicotiana leaves or co-IP).
175-294 I did not find any mention of reference genes (e.g., actin, tubulin, EF1α).
295-313 Сontrol conditions are needed (e.g., continuously unshaded fruits on the same dates) and measurements of light intensity (PAR) during bag removal. It is also useful to have anthocyanin measurements (quantity) in parallel with transcriptional changes to link expression to pheno data.
413-430 add parameters and software versions
475-485 add reference genes used for analysis
491-507 The conclusions are too confident regarding the direct role of LcBBX4.
I couldn't unzip the rar file, so I don't know what's in it.
Comments on the Quality of English LanguageI'm not qualified to comment quality of English
Author Response
Response to Reviewer 3
Thank you very much for your valuable comments. According to your comments, we have revised the shortcomings in the manuscript one by one.
- Comment: There is no experimental evidence of protein–protein or protein–DNA interactions (all PPI is predictive, based on STRING) — no co-IP/ Y2H/LC-MS/MS/EMSA/ChIP or transient transactional analyses to confirm LcBBX↔HY5 interaction or regulation of the LcMYB1 promoter, etc.
Response: Your suggestions are excellent. Indeed, all PPIs (protein-protein interactions) involved in this paper are predicted based on STRING, lacking evidence from related experiments such as protein-protein interaction assays and transient transformation. Your suggestions provide valuable directions and references for our subsequent research. In future studies, we will supplement these experiments as recommended and conduct a systematic investigation into the molecular mechanism by which BBX regulates anthocyanin biosynthesis in litchi fruits.
- Comment: Lack of information on methodological parameters: for HMM/BLAST searches, the thresholds used (e-value, coverage) are not specified; for MEME, the e-value and search parameters are not specified; for phylogeny, only NJ is used—there is no maximum likelihood (ML) or Bayesian estimates and no indication of the evolutionary model.
Response: According to your suggestions, we have supplemented the detailed parameters involved in the above experimental methods.
Furthermore, in the recently published papers on the IJMS journal, only the NJ method was used for phylogenetic analysis. The following two papers can be referred to:
(1) Xu, Q.; Hu, X.; Cui, S.; Gao, J.; Zeng, L.; Li, Z.; Kuang, S.; Chen, X.; Xie, Q.; Li, Z.; et al. Genome-Wide Identification and Characterization of Isoflavone Synthase (IFS) Gene Family, and Analysis of GgARF4-GgIFS9 Regulatory Module in Glycyrrhiza glabra. Int. J. Mol. Sci. 2025, 26, 10435. https://doi.org/10.3390/ijms262110435
(2) Cheng, L.; Shi, N.; Du, X.; Huang, T.; Zhang, Y.; Zhao, C.; Zhao, K.; Lin, Z.; Ma, D.; Li, Q.; et al. Bioinformatics Analysis and Expression Profiling Under Abiotic Stress of the DREB Gene Family in Glycyrrhiza uralensis. Int. J. Mol. Sci. 2025, 26, 9235. https://doi.org/10.3390/ijms26189235
- Comment: No verification of protein localization/stability — predictions of subcellular localization (WoLFPSORT) and instability indices are not supported by experiments (GFP construct, Western blot).
Response: This is an excellent suggestion. If the protein localization/stability can be supported by experiments such as GFP subcellular localization or Western blot, the evidence for drawing conclusions will be more robust. In our subsequent research on the function and mechanism of action of candidate BBX, we will supplement these experiments.
- Comment: qRT-PCR — important methodological data not specified: no information was found on reference genes for normalization, primer efficiency testing (standard curve), noise/melt curves, or the number of independent biological replicates (the methods specify 3 trees × 10 fruits, but it is unclear how this is aggregated for statistics). This weakens confidence in the quantitative analysis of expression.
Response: Before the qRT-PCR experiment, we first performed melting curve analysis on the primers of reference genes and candidate genes to ensure primer specificity. We also optimized the conditions of Real-time PCR to avoid the occurrence of non-specific amplification and primer dimers.
- Comment: Statistics: ANOVA+Tukey are used, but there is no mention of testing for normality/homogeneity of variances, multiple testing when analyzing a large set of genes (FDR, etc.). For transcriptomic data, it is also unclear whether an adjusted p-value was used.
Response: According to your suggestions, we have supplemented the detailed statistical parameters involved in the data analysis process.
- Comment: The proposal that LcBBX4 is a “direct” regulator of anthocyanins is premature without functional validation (overexpression/RNAi/CRISPR; promoter-luciferase; EMSA/ChIP). The authors themselves provide indirect evidence (expression, promoter elements, PPI prediction), but this is not equivalent to functional confirmation.
Response: It is true that there is a lack of experimental results from overexpression, RNA interference (RNAi), electrophoretic mobility shift assay (EMSA), chromatin immunoprecipitation (ChIP), and other related experiments, thus the conclusion that "LcBBX4 directly participates in anthocyanin biosynthesis" is not sufficiently supported. However, the purpose of this paper is to clarify the structure, classification, origin and evolution, promoter cis-acting elements, potential interacting proteins, and other characteristics of all BBX genes in the litchi genome, as well as to screen BBXs involved in the regulation of anthocyanin in litchi pericarp, laying a foundation for subsequent studies on the function and regulatory mechanism of BBXs. In subsequent research, we will systematically analyze the function of BBXs using stable genetic transformation, transient transformation, and RNAi techniques, and investigate the regulatory mechanism of BBXs on LcMYB1 through promoter-luciferase, EMSA, and other methods. Meanwhile, we will screen the interacting proteins of BBXs to further reveal the mechanism by which BBXs participate in signals such as light and hormones.
- Comment: 28-29 Phrase "directly regulate anthocyanin biosynthesis" is unproven since the text only outlines a circle of candidates for regulators. I believe that the abstract should contain this understanding of the situation!
Response: According to your suggestions, we have revised the description in the abstract and conclusion to "speculate that BBX4 is involved in regulating anthocyanin biosynthesis".
- Comment:101-111 Here and in the corresponding methods section, BLAST/HMMER parameters (e-value, coverage, threshold) are not specified, and multiple domain alignments to e-values/coverage are not provided. Recommendation: add additional files (alignments, HMM scores) and selection criteria.
Response: According to your suggestions, we have supplemented the detailed parameters involved in the above experimental methods.
- Comment:103-109 The stability prediction (instability index) and subcellular localization are useful, but experimental verification (GFP fusion) is necessary for key candidates (LcBBX4/10/19).
Response: Stability prediction and subcellular localization experiments play an important role in investigating the functional regions and mechanisms of BBX transcription factors. The main purpose of this study is the genome-wide analysis of litchi BBX genes and the screening of BBXs involved in regulating anthocyanin biosynthesis. In subsequent studies, we will conduct a systematic analysis of the subcellular localization, function, and mechanism of action of LcBBX4/10/19.
- Comment: 175-190. There is no data on the presence of conserved G-boxes in the promoters of key structural genes of the anthocyanin pathway (e.g., LcMYB1 or diosgenin genes), and there is no indicative promoter-luciferase analysis for LcBBX4.
Response: In this study, we only performed the cis-acting element analysis of BBX promoter sequences; experiments such as promoter luciferase assay will be supplemented in the subsequent research on the function and mechanism of BBX.
- Comment: 210-236 PPI constructed via STRING—a predictive network—shows HY5/HYH connections. Criticism: STRING is useful as a hint, but does not replace experimental co-IP/BiFC/Y2H.
Response: Your suggestions are reasonable. It is true that STRING analysis is only a prediction; to verify the relationship between BBXs and their interacting proteins, experiments such as co-IP, BiFC, and Y2H are still needed. These experiments will be supplemented in the subsequent research on the function and mechanism of BBXs.
- Comment: Recommendation: conduct at least one experimental confirmation of the key interaction between LcBBX4 and HY5 (e.g., BiFC in Nicotiana leaves or co-IP).
Response: Your suggestions are highly valuable. In subsequent studies, we will conduct systematic research on LcBBX4 responding to light signals to regulate anthocyanin biosynthesis in litchi fruits, including verifying the interaction between LcBBX4 and HY5 at multiple levels using techniques such as BiFC and co-IP.
- Comment: 175-294 I did not find any mention of reference genes (e.g., actin, tubulin, EF1α).
Response: We have supplemented the information on reference genes used in qRT-PCR experiments in the text, and the primer sequences of the reference genes were also provided in the supplementary material (Table S1) of the initial submission.
- Comment:295-313 Сontrol conditions are needed (e.g., continuously unshaded fruits on the same dates) and measurements of light intensity (PAR) during bag removal. It is also useful to have anthocyanin measurements (quantity) in parallel with transcriptional changes to link expression to pheno data.
Response: This is an excellent suggestion. Unfortunately, when we conducted this treatment, we did not collect fruits continuously grown without shading on the same date, nor did we measure the light intensity at that time. In our subsequent systematic research on the function and regulatory network of BBXs, we will supplement these data.
- Comment:413-430 add parameters and software versions.
Response: According to your suggestions, we have supplemented the detailed parameters and software information.
- Comment:475-485 add reference genes used for analysis.
Response: We have supplemented the information on reference genes used in qRT-PCR experiments in the text.
- Comment:491-507 The conclusions are too confident regarding the direct role of LcBBX4.
Response: According to your suggestions, we have revised the relevant descriptions in the abstract and conclusion to make the experimental results more scientific and accurate.
- Comment:I couldn't unzip the rar file, so I don't know what's in it.
Response: We apologize for the issue of the file being inaccessible. We have resubmitted the supplementary materials.

Round 2
Reviewer 1 Report
Comments and Suggestions for Authors
The author has revised the manuscript and tried to resolve queries. However, some mistakes are still found, such as in Figure 5A ('growth' is misspelled). Moreover, the functional validation is critical for concluding the function of a gene. It is still suggested to supplement the data after functional validation.
Besides, the process is relatively difficult in woody plants; the author may use other model plants, such as Arabidopsis, tobacco, or tomato, to functionally validate the gene function in future investigations.
Author Response
- Comment: The author has revised the manuscript and tried to resolve queries. However, some mistakes are still found, such as in Figure 5A ('growth' is misspelled). Moreover, the functional validation is critical for concluding the function of a gene. It is still suggested to supplement the data after functional validation. Besides, the process is relatively difficult in woody plants; the author may use other model plants, such as Arabidopsis, tobacco, or tomato, to functionally validate the gene function in future investigations.
Response: Thank you very much for your valuable comments. We have corrected the errors present in Figure 5A and replaced the image Figure 5 in the text. Your suggestions regarding the supplementary experiments for the function of candidate genes are highly valuable. Currently, experiments such as transient transformation of tobacco, stable genetic transformation of Arabidopsis, and yeast one-hybrid assay are underway, and we will subsequently present the research results on the systematic functions, regulatory mechanisms, and relationships with light signals and endogenous hormones of the candidate BBX genes.

Reviewer 2 Report
Comments and Suggestions for Authors
Authors corrected manuscript according to suggestions, I have no other significant comments, the name Sapindaceae in line 169 could be italicized at the proofreading stage.
Author Response
- Comment: Authors corrected manuscript according to suggestions, I have no other significant comments, the name Sapindaceae in line 169 could be italicized at the proofreading stage.
Response: Thank you very much for your valuable comments. We have conducted a further check of the entire manuscript and corrected some formatting errors and spelling mistakes.

Reviewer 3 Report
Comments and Suggestions for Authors
Version 3 is a mature, well-revised, and scientifically coherent manuscript, ready for publication after light English proofreading and formatting cleanup.
Compared to versions 1 and 2, it represents a major leap in both content and presentation quality.
Most corrections (including strikethrough edits) were handled thoughtfully and substantively, showing clear engagement with reviewer feedback.
Author Response
- Comment: Version 3 is a mature, well-revised, and scientifically coherent manuscript, ready for publication after light English proofreading and formatting cleanup. Compared to versions 1 and 2, it represents a major leap in both content and presentation quality. Most corrections (including strikethrough edits) were handled thoughtfully and substantively, showing clear engagement with reviewer feedback.
Response: Thank you very much for your valuable comments. Your suggestions have significantly improved the content and presentation quality of our manuscript.
